# Prevalence and Risk Factors of Adverse Effects and Allergic Reactions after COVID-19 Vaccines in a Mexican Population: An Analytical Cross-Sectional Study

**DOI:** 10.3390/vaccines10122012

**Published:** 2022-11-25

**Authors:** Jesús Maximiliano Granados Villalpando, Sergio de Jesus Romero Tapia, Guadalupe del Carmen Baeza Flores, Jorge Luis Ble Castillo, Isela Esther Juarez Rojop, Frida Isabel Lopez Junco, Viridiana Olvera Hernández, Sergio Quiroz Gomez, Jesús Arturo Ruiz Quiñones, Crystell Guadalupe Guzmán Priego

**Affiliations:** 1Cardiometabolism Laboratory, Research Center, Academic Division of Health Sciences (DACS), Juarez Autonomous University of Tabasco (UJAT), Villahermosa 86040, Mexico; 2Health Sciences Academic Division (DACS), Juarez Autonomous University of Tabasco (UJAT), Villahermosa 86040, Mexico; 3Metabolic Disease Biochemistry Laboratory, Research Center, Academic Division of Health Sciences (DACS), Juarez Autonomous University of Tabasco (UJAT), Villahermosa 86040, Mexico; 4Lipid Metabolism Laboratory, Research Center, Academic Division of Health Sciences (DACS), Juarez Autonomous University of Tabasco (UJAT), Villahermosa 86040, Mexico; 5Research Center for Tropical and Emerging Diseases, High Specialty Regional Hospital “Dr. Juan Graham Casasús”, Villahermosa 86126, Mexico

**Keywords:** adverse effects, allergic reactions, COVID-19, vaccination, Mexico

## Abstract

Vaccinations have helped to control the COVID-19 pandemic; however, few studies focus on the adverse effects and allergic reactions of these vaccines and fewer have a scope in the Latin American population. The objective of this study was to assess the associations between vaccinations, sex, age, allergic reactions, and adverse effects. This was an analytical cross-sectional study conducted between 1 July and 1 October 2022. The sample consisted of 443 surveyed participants, with a total of 1272 COVID-19 vaccine doses. Seven vaccines (Pfizer BioNTech, Oxford-AstraZeneca, CanSino, Moderna, Johnson and Johnson, Sinovac, and Sputnik V) were evaluated. A total of 12.6% of those surveyed had at least one allergic reaction posterior to vaccination, and females had a greater chance of developing one (*p* < 0.001, OR 3.1). The most common allergic reaction was chest pain, and Pfizer-BioNTech and Oxford-AstraZeneca were associated with the onset of allergic reactions (*p* < 0.005). A total of 54.6% of those surveyed developed adverse effects, the most common of which were myalgia, fever, cephalea, asthenia or adynamia, and arthralgia; moreover, older age was associated with the onset of adverse effects (*p* < 0.5). This study concludes that the BNT162b2 (Pfizer BioNTech) and ChAdOX1 nCOV-19 (Oxford-AstraZeneca) vaccines are strongly associated with the onset of allergic reactions, with ORs of 1.6 (CI 95%, 1.18 to 2.3) and 1.87 (CI 95%, 1.35 to 2.6), respectively. In addition, females have a greater chance of developing allergic reactions associated with COVID-19 vaccinations, and there was a relation found between older age and a greater prevalence of comorbidities, adverse effects after vaccination, and COVID-19 infection after vaccination.

## 1. Introduction

COVID-19 is an infectious disease caused by the SARS-CoV-2 virus that emerged in Wuhan, China in December 2019. On 11 March 2020, WHO determined COVID-19 to be a pandemic due to the alarming propagation and severity of the disease Continuous research has yielded the development of new forms of prevention, such as vaccines. Some of the most important platforms used due to their potential are those based in recombinant subunits against SARS-CoV-2 [1,2,3,4,5,6,7].

Different types of COVID-19 vaccines have been utilized and developed around the world, some of which are based on mRNA, such as BNT162b2 (Pfizer BioNTech, Brooklin, New York, United States) and mRNA-1273 (Moderna, Cambridge, Massachusetts, United States), with a low security risk and high efficacy (95% and 94.1%, respectively). There also exists vector vaccines, such as ChAdOX1 nCOV-19 (Oxford-AstraZeneca, England, United Kingdom), Ad26.COV2. S (Johnson and Johnson, New Brunswick, New Jersey, United States), Gam-COVID-Vac (Sputnik V, Moscow, Russia), and Ad5-nCoV-S (CanSino, Tianjin, China), which have 70.4%, 66%, 92%, and 91.6% efficacies, respectively. Inactivated vaccines, such as CoronaVac (Sinovac), are also actively being used in some countries such as Mexico, although they are not so effective (60%) [7,8,9,10].

Epidemiological surveillance and clinical follow-ups of “new” COVID-19 vaccines are essential and indispensable to facilitate the detection, research, and analysis of events supposedly attributable to vaccination or immunization (ESAVI). Such ESAVI are generally adverse effects that can be potentially capable of threatening life [11].

Adverse effects are harmful events that are caused by or related to medication. Among all adverse drug reactions, those that are related to hypersensitivity or immunological mechanisms are known as allergic reactions [12,13]. Several studies have been conducted to identify the possible adverse effects associated with COVID-19 vaccines which can be present in up to 88% of cases, mainly involving pain at the injection site, fatigue, cephalea, and myalgia [13,14,15].

In a comparative analysis of the biological and pharmacological characteristics and adverse reactions/side effects of BioNTech and Moderna vaccines, the main allergic reactions included hives and non-frequently anaphylaxis, and the main adverse effects included myalgia, pain at the puncture site, fever, drowsiness, arthralgia, emesis, chills, and fatigue [15,16,17].

In previous studies, it has been demonstrated that female subjects, comorbid subjects, subjects younger than 50 years old, and those with a history of allergies have a greater chance of developing more and stronger adverse effects [18,19]. The main purpose of this study is to assess the associations between different vaccines applied in Mexico and the adverse effects and/or allergic reactions.

## 2. Materials and Methods

### 2.1. Study Design and Data Sources

The present analytical cross-sectional study was designed to have a scope regarding the adverse effects and allergic reactions associated with COVID-19 vaccinations in a Latin American population, specifically a Mexican population. Seven vaccines were evaluated: Pfizer-BioNTech (BNT162b2); Oxford-AstraZeneca (ChAdOx1-S); CanSino (Ad5-nCoV-S); Moderna (mRNA-1273); Johnson and Johnson (Ad26.COV2. S); Sinovac (CoronaVac); and Sputnik V (Gam-COVID-Vac). The presentation of adverse effects and allergic reactions, as well as sex, the development of COVID-19 posterior to vaccination, comorbidities (with a focus in atopic diseases), and age were considered. The STROBE guidelines for cross-sectional studies were used to hereby write the paper [20].

### 2.2. Study Setting

This study was conducted from 1 July 2022 to 1 October 2022, during the fifth COVID-19 wave, via an online survey distributed to various inhabitants of Tabasco, Mexico.

### 2.3. Participants and Procedures

The participants in the study were a sample calculated from a total of 81,177,465 subjects (as of 4 July 2022) vaccinated in Mexico with at least one COVID-19 vaccine, and with a 95% confidence interval, a 5% margin of error, and a proportion of 50%, consisting of volunteer-targeted sampling. To be eligible for the study, all those surveyed had to provide informed consent. Both sexes were admitted, their age had to be between 18 and 65 years old, only inhabitants of Tabasco, Mexico were accepted, and the participants had to answer all sections of the survey. Lactating and pregnant women were excluded. Those surveyed were not compensated for their participation in this study [21].

A total of 385 subjects were targeted for study, 460 subjects answered the survey, and 12 participants were excluded because of age (less than 18 years old), and 5 participants were excluded because of pregnancy, resulting in a final sample of 443 subjects; therefore, the margin of error was reduced to 4.66%.

### 2.4. Measures, Variables, and Data Collection

All participants answered an online survey consisting of general data and an exploratory questionnaire designed by the Cardiometabolism Laboratory in the Research Center of the Academic Division of Health Sciences of the Juarez Autonomous University of Tabasco. The survey was validated via the Delphi method with the participation of five experts in this type of study, including two medical doctors who specialized in allergology, internal medicine. and infectiology, a medical doctor and two pharmacists with PhDs in Biomedical Sciences, and the final three belonged to the National System of Researchers. The Delphi method consisted of two rounds in which a consensus was achieved.

This survey was designed for typification and epidemiological classification, including sex, age, vaccines applied, the development of COVID-19 posterior to vaccination, comorbidities (with a focus on atopic diseases), number of doses, the development of adverse effects, specific adverse effects, the development of allergic reactions, and specific allergic reactions.

To assess the prevalence of comorbidities, seven different general comorbidities were considered, including diabetes, hypertension, thyroid disorders, rheumatological disorders, cancerous diseases, overweight and obesity, and atopic diseases. Atopic diseases, due to being closely related to immunological and hypersensitive responses, were further assessed as food allergies, drug allergies, atopic dermatitis, allergic rhinitis, and asthma. Allergic reactions were considered as the following: thoracic pain; dyspnea; hives; anaphylaxis; angioedema; timelapse between vaccination and symptoms onset; and whether medical attention was required or not. Adverse effects were considered as the following: myalgia; arthralgia; fever; cephalea; asthenia and adynamia; drowsiness; shivering and chills; pain at the puncture site; nasal discharge; hypotension; nausea; cough; and diarrhea.

### 2.5. Statistical Analysis

Data were analyzed using Microsoft Excel (2021 version), Prism (Version 9; GraphPad), and SPSS (v26), the software licenses used were provided by the information technology department of the Juarez Autonomous University of Tabasco. Qualitative variables were assessed via a Chi-squared test, Fisher’s exact test, and OR (odds ratio) were calculated. The continuous variables were assessed for normality using the Kolmogorov–Smirnov test and they all showed normal distribution (*p* < 0005). Due to this, a parametric inferential test was conducted, and a Student’s T-test was used to analyze group differences between the variables previously mentioned. A *p*-value of 0.05 was considered statistically significant.

## 3. Results

The sample consisted of 443 surveyed participants, of which 302 (68.2%) were females and 141 (31.8%) were males, aging from 18 to 68 years old. Almost a quarter of all those surveyed had at least one comorbidity, and atopic diseases and overweight and obesity were the most common ones, followed by hypertension. A Fisher’s exact test was utilized to assess the hypothesis that distribution leaned towards the female sex and the risk was assessed via OR (odds ratio) (Table 1). The mean age of all those surveyed was 25.93 (SD of ±11.29).

Of the 56 of those surveyed who had atopic diseases, 19 had drug allergies, 14 had food allergies, 1 had atopic dermatitis, 8 had allergic rhinitis, and 35 had asthma. More than half had more than one atopic disease.

After assessing normality in the distribution via a Kolmogorov–Smirnov Test (*p* < 0.001), a Student’s T-test was utilized to assess the differences in ages between the prevalence of comorbidities, adverse effects, and allergic reactions. There was a significant difference between ages and the prevalence of comorbidities (*p* < 0.0001), COVID-19 after vaccination (*p* < 0.05), and adverse effects (*p* < 0.05) after vaccination. There was no statistically significant difference between age and allergic reactions nor between comorbidities and adverse effects or allergic reactions via Chi-squared test.

A total of 1272 doses of any COVID-19 vaccine were applied in 443 subjects, all of whom had at least one dose. A total of 435 (98.2%) had at least two doses, 339 (78.8%) had at least three doses, and 45 (12.4%) had all four doses of any COVID-19 vaccine.

Of the 1272 applied doses, 50.62% (644) were Pfizer-BioNTech, 41.9% (533) were Oxford-AstraZeneca, 5.03% (64) were CanSino, 1.2% (16) were Moderna, 0.07% (1) was Johnson and Johnson, 0.94% (12) were Sinovac, and 0.15% (2) were Sputnik V. There was a total of 166 (13.05%) doses that caused allergic reactions in 57 (12.87%) of the subjects (Table 2).

The allergic reaction doses were presented as follows: In all cases, there were at least two allergic reactions. The most common allergic reaction was chest pain with 123 (74.1%) incidences, followed by dyspnea with 101 (60.8%), hives with 18 (10.8%), and anaphylaxis and angioedema with 9 (5.4%). A total of 100% of the allergic reactions began in the first 60 min after vaccination. Only three of those surveyed with allergic reactions required medical attention, and those three received AstraZeneca vaccines (Table 3).

Of the 1272 vaccine doses, 727 (57.15%) caused adverse effects (Table 4) in 242 (54.62%) subjects. The most common adverse effects in all 443 of those surveyed were myalgia and fever (30.47%), asthenia or adynamia (25.7%), cephalea (25%), and arthralgia (24.8%). (Figure 1).

Although with a *p* value of 0.06, comorbidities had a high OR of 7.63 (CI 95%, 0.63 to 92.15) in the context of the necessity of medical attention after vaccination. The same situation (no statistically significant value but a high OR) occurred with atopic diseases and anaphylaxis after vaccination (OR of 2.4, CI 95%, 0.38 to 14.88), as well as the necessity of medical attention after vaccination (OR of 2.8, CI 95%, 0.22 to 34.79), drug allergies and anaphylaxis after vaccination (OR of 7.83, CI 95%, 0.431 to 142.23), food allergies and anaphylaxis (OR of 3.9, CI 95%, 0.3 to 49.98), allergic reactions (OR of 2.36, CI 95%, 0.62 to 9.01), and asthma and chest pain after vaccination (OR of 2.1, CI 95%, 0.22 to 21.02).

## 4. Discussion

The present study has demonstrated that there is a relation between BNT162b2 (Pfizer-BioNTech) and ChAdOx1-S (Oxford-AstraZeneca) with allergic reactions, with ORs of 1.6 (CI 95%, 1.18 to 2.3) and 1.87 (CI 95%, 1.35 to 2.6), respectively. This shows that, although there is a high efficacy and security in both vaccines, they are greatly associated (with statistical significance) with allergic reactions, such as chest pain, dyspnea, angioedema, hives, and anaphylaxis. This is consistent with other studies that name several adverse effects and allergic reactions associated with BNT162b2 and ChAdOx1-s [22,23,24,25,26,27].

There is also a statistically significant relationship between the female sex and thyroid diseases (which has been described) and allergic reactions associated with COVID-19 vaccines, which prior studies have demonstrated as “worst side effects” or “more severe adverse events” [22,23,24].

This study is consistent with other studies that confirm there is a higher prevalence of side effects and more severe adverse effects in women than in men, meaning there is a sex component in this variable [23,24].

In contrast, this study opposes what other studies have discovered about age and side effects. In most studies, an association between younger age and a higher prevalence of side effects/adverse effects has been found; however, in this study, older age seems to be more associated with a higher prevalence of adverse effects, though this could be a confounding variable that will be assessed in further paragraphs [19,24].

Allergic reactions were present in 166 (13.05%) doses and were always accompanied by another allergic reaction. Additionally, the most common allergic reaction in all vaccines was chest pain which was present in 74.1% of cases, closely followed by dyspnea (60.8%). Only 1.8% of allergic reactions required medical attention, and they all were due to an Oxford-AstraZeneca vaccination. Adverse effects were present in more than half of the 1272 doses (727, 57.15%). The most common adverse effects in all vaccines were myalgia (90.5%) and fever (85.7%).

This work highlights the importance of surveillance during the application of COVID-19 vaccines. Despite the fact that only 12.6% of those surveyed had allergic reactions and only three required medical attention, this is in no way a small number, exhorting the clinicians to carefully treat and prevent these allergic reactions.

Additionally, this study stresses the relevance of the age of patients as there was a relation between age and comorbidities, COVID-19 after vaccination, and adverse effects. Furthermore, it marks a relatively high prevalence of asthma (7.8%) in this context, which should be assessed in other studies.

The strengths in this study can be summarized by the statistically significant outcomes in the case of the association between the female sex and allergic reactions, the association between the Pfizer-BioNTech and Oxford-AstraZeneca vaccines and allergic reactions, and the association between age, comorbidities, and COVID-19 after vaccination and adverse effects.

Nevertheless, there are several weaknesses and opportunities in this study. The sample, although calculated, has a relatively high margin of error (4.66%) and in order to correct it, the sample must increase its numbers. It is also limited by what the survey participants (who are generally not clinicians) can understand. Furthermore, the study is cross-sectional; therefore, the answers were linked to what the survey participants could remember about their vaccination and booster experiences. Finally, most of the survey participants were young and therefore had a better immune system and less comorbidities. A better sex and age distribution is desirable.

There is a confounding variable in the number of allergic reactions related to the different vaccines, which can be interpreted as safeness; however, this could be linked to the lack of doses in this context, mainly considering that 1177 (92.5%) of the 1272 doses were either from the Pfizer-BioNTech or Oxford-AstraZeneca vaccines, and they were coincidentally the vaccines associated with allergic reactions; therefore, it should be considered that this might be the case because they are the most used in this context.

Another confounding variable that was previously mentioned is age, which was associated with a greater chance of developing adverse effects; nevertheless, it shall be considered that, in this study, older subjects had a higher prevalence of comorbidities, and this might be the reason why older subjects were associated with a higher incidence of adverse effects, masking the real reason which is comorbidities.

Similar works have been conducted in other countries which assess the side effects in BNT162b2 and ChAdOx1-s vaccines, and most of them have had consistent results in that both vaccines had a relatively high, or at least modest, incidence of side effects [28,29,30,31,32,33,34,35].

ChAdOx1-s has been demonstrated to have a high incidence of side effects, mainly injection site pain, fatigue, chills, and muscle pain, subsiding in the first three days after vaccination [35,36]. BNT162b2 has also been demonstrated to frequently produce mild side effects, mainly injection site pain, fatigue, and muscle pain, predominantly in females [25,36]. To improve the significance of studies such as this one, we recommend that a larger cohort study with a lower margin of error is taken up in future vaccinations and studies, expanding the database with more variables, such as infection prior to vaccination, the severity of COVID-19 infection prior to vaccination, vaccine mixing (heterologous boosters), the order of heterologous boosters, the use of drugs to lessen adverse effects, the use of drugs to lessen allergic reactions, and prior allergic reactions to other vaccinations. Additionally, the participation of multiple centers and a longitudinal design to assess adverse events associated with vaccinations, as well as a better distribution between age groups, vaccines used, and sexes, should be beneficial to posterior studies.

## 5. Conclusions

After almost three years since the first COVID-19 case, COVID-19 vaccines have utterly demonstrated their efficacy all around the world. This study hereby presents data from a Mexican population and emphasizes what has already been confirmed in other vaccine studies: females are more prone to allergic reactions from vaccinations than males by a more than two hundred percent chance; and the older the person is, the greater the chance of developing adverse effects is.

This study denotes that although Pfizer-BioNTech and Oxford-AstraZeneca are effective, they are also statistically associated with allergic reactions after vaccination in comparison with all other vaccines, having a relatively high prevalence and incidence.

## Figures and Tables

**Figure 1 vaccines-10-02012-f001:**
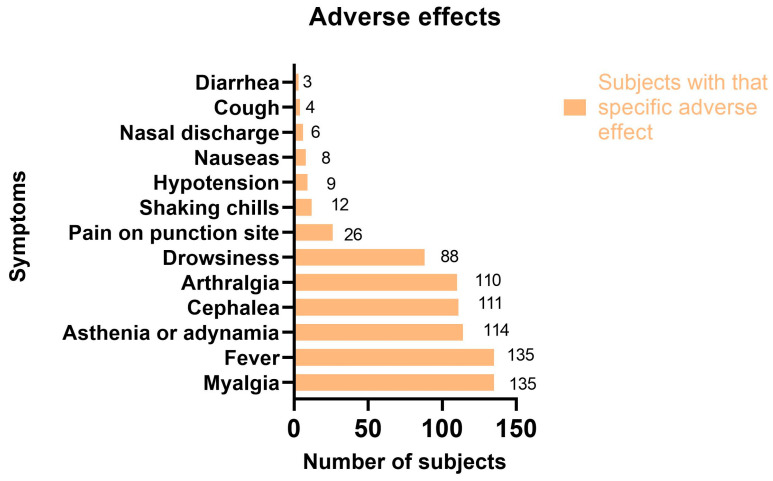
Adverse effects in surveyed participants.

**Table 1 vaccines-10-02012-t001:** Description of sample by sex.

Sex	Males, *n* = 141 (31.80%)	Females, *n* = 302 (68.2%)	All, *n* = 443 (100%)	100%	
Age #	26.4 (±12.06)	25.72 (±10.92)	25.93 (±11.29)		
	*n* (%)	*n* (%)	*n* (%)	*p* Value	
Comorbidities	39 (37.2)	66 (62.8)	105 (23.7)	0.1	0.7 (0.46 to 1.15)
Hypertension	6 (50)	6 (50)	12 (2.7)	0.1	0.4 (0.14 to 1.44)
Diabetes	3 (37.5)	5 (62.5)	8 (1.8)	0.4	0.7 (0.18 to 3.28)
Overweight or obesity	17 (33.3)	36 (66.6)	54 (12.1)	0.5	0.9 (0.53 to 1.84)
Thyroidal disorders	0 (0)	8 (100)	8 (1.8)	<0.05 *	1 (1 to 1.04)
Cancer	1 (100)	0 (0)	1 (0.2)	0.3	0.9 (0.97 to 1)
Rheumatological	0 (0)	1 (100)	1 (0.2)	0.4	1 (0.99 to 1.01)
Atopic disease	23 (41.1)	33 (58.9)	56 (12.6)	0.07	0.6 (0.35 to 1.12)
COVID-19 posterior to vaccination	50 (35.5)	91 (64.5)	141 (31.8)	0.2	0.7 (0.51 to 1.19)
Allergic reactions	8 (14.3)	48 (85.7)	56 (12.6)	<0.001 **	3.1 (1.44 to 6.83)
Adverse effects	83 (34.3)	159 (65.7)	242 (54.6)	0.1	0.7 (0.51 to 1.16)

# number, * *p* value < 0.05, ** *p* value < 0.005.

**Table 2 vaccines-10-02012-t002:** Total vaccines doses and allergic reactions.

Vaccine Doses	No Allergic Reactions, *n* (%)	Allergic Reactions, *n* (%)	Fisher’s Test *p* Value	Odds Ratio (CI 95%)
Pfizer-BioNTech	578 (45.4)	66 (5.2)	0.003 **	1.6 (1.18 to 2.3)
Other vaccines	528 (41.5)	100 (7.8)		
Oxford-AstraZeneca	441 (34.6)	92 (7.2)	0.002 **	1.87 (1.35 to 2.6)
Other vaccines	665 (52.3)	74 (5.8)		
CanSino	59 (4)	5 (0.4)	0.4	0.7 (0.27 to 1.64)
Other vaccines	1047 (82.9)	161 (12.6)		
Moderna	15(1)	1 (0.07)	0.7	2.2 (0.38 to 24.15)
Other vaccines	1091 (86.3)	165 (13)		
Johnson and Johnson	1 (0.07)	0 (0)	1	Not enough subjects
Other vaccines	1105 (86.9)	166 (13.05)		
Sinovac	11 (0.086)	1 (0.07)	1	1.6 (0.29 to 18.01)
Other vaccines	1095 (86)	165 (13)		
Sputnik V	1 (0.07)	1 (0.07)	0.2	0.14 (0 to 2.8)
Other vaccines	1105 (87)	165 (13)		

** *p* value < 0.005.

**Table 3 vaccines-10-02012-t003:** Allergic reactions and vaccines doses.

Vaccines	Total Doses That Caused Allergic Reactions	Chest Pain	Dyspnea	Hives	Anaphylaxis	Angioedema	Required Medical Attention
Pfizer-BioNTech	66	40 (60.6%)	35 (53%)	8 (12.12%)	4 (6%)	1 (1.5%)	0
Oxford-AstraZeneca	92	75 (81.5%)	60 (65.2%)	8 (8.7%)	5 (5.4%)	6 (6.5%)	3 (3.3%)
CanSino	5	5 (100%)	4 (80%)	3 (60%)	0	1 (100%)	0
Moderna	1	1 (100%)	0	1 (100%)	0	1 (100%)	0
Johnson and Johnson	0	0	0	0	0	0	0
Sinovac	1	1 (100%)	1 (100%)	1 (100%)	0	0	0
Sputnik V	1	1 (100%)	1 (100%)	0	0	0	0
All vaccines	166	123 (74.1%)	101 (60.8%)	18 (10.8%)	9 (5.4%)	9 (5.4%)	3 (1.8%)

**Table 4 vaccines-10-02012-t004:** Adverse effects per vaccine doses.

	All Vaccines	Pfizer-BioNTech	AstraZeneca	CanSino	Moderna	Johnson and Johnson	Sinovac	Sputnik V
Total adverse effects (% of all doses)	727	329 (45.25%)	334 (45.94%)	50 (6.8%)	12 (1.65%)	0	1 (0.13%)	1 (0.13%)
Myalgia	658 (90.5%)	300 (91.1%)	306 (91.6%)	42 (84%)	8 (66.6%)	0	1 (100%)	1 (100%)
Fever	623 (85.7%)	278 (84.5%)	300 (89.8%)	40 (80%)	4 (33.3%)	0	1 (100%)	0
Asthenia or adynamia	180 (24.7)	78 (23.7%)	60 (17.9%)	34 (68%)	6 (50%)	0	1 (100%)	1 (100%)
Cephalea	71 (9.7%)	20 (6%)	35 (10.4%)	10 (20%)	4 (33.3%)	0	1 (100%)	1 (100%)
Arthralgia	120 (16.5%)	47 (14.2%)	50 (14.9%)	18 (36%)	4 (33.3%)	0	1 (100%)	0
Drowsiness	119 (16.3%)	60 (18.2%)	45 (13.4%)	12 (24%)	1 (8.3%)	0	1 (100%)	0
Pain at puncture site	142 (19.5%)	57 (17.3%)	65 (19.4%)	15 (30%)	3 (25%)	0	1 (100%)	1 (100%)
Shaking chills	116 (15.9%)	52 (15.8%)	63 (18.8%)	1 (2%)	0	0	0	0
Hypotension	11 (1.5%)	5 (1.5%)	4 (1.2%)	2 (4%)	0	0	0	0
Nauseas	10 (1.3%	6 (1.8%)	2 (0.6%)	2 (4%)	0	0	0	0
Nasal discharge	9 (1.2%)	4 (1.2%)	3 (0.9%)	0	2 (16.6%)	0	0	0
Cough	128 (1.6%)	3 (0.9%)	3 (0.9%)	4 (8%)	2 (0.6%)	0	0	0
Diarrhea	4 (0.5%)	2 (0.6%)	2 (0.6%)	0	0	0	0	0

## Data Availability

The datasets used and/or analyzed in the current study are available from the corresponding author on reasonable request.

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
