# Peer review of "Prevalence and Risk Factors of Adverse Effects and Allergic Reactions after COVID-19 Vaccines in a Mexican Population: An Analytical Cross-Sectional Study"

_vaccines, 2022, doi:10.3390/vaccines10122012_

Round 1

Reviewer 1 Report

The authors present an observational, prospective analytical cross-sectional study conducted during July 1st to October 1st of 2022. The sample consisted of 443 surveyed and seven vaccines (Pfizer BioNTech, Oxford-AstraZeneca, CanSino, Moderna, 23 Johnson & Johnson, Sinovac and Sputnik V) were evaluated.

The study is scientifically interesting and well written, nevertheless the authors need to address some critical points.
Pag 4

This part should be rewritten. The total number of doses is confusing.

The sample is small and there is an imbalance between men and women in the sample. The authors should think of identifying a larger cohort of patients, with a better distribution between men and women and also a greater homogeneity of age. And divide the sample by vaccines and analyze adverse events and side effects separately 

Author Response

I attach the document with the responses to the observations.

Reviewer 2 Report

Dear author(s),

Thank you for your esteemed efforts in increasing our collective knowledge of COVID-19 vaccine safety.

Please consider the following points:

Title

1. Add the study design to the title.

2. What is the difference between adverse events and side effects? You did not really focus on this point in your study. Therefore I suggest using a single term here.

3. You can add the words "Prevalence and Risk Factors" at the beginning of your title to show the purpose of your work.

Abstract

4. Line 19-20: please fix this sentence. It is unclear.

5. Line 21: cross-sectional studies can not be described as prospective or retrospective because they have no follow-up component. Remove the word prospective".

6. Line 21: cross-sectional studies belong to observational epidemiology. There is no benefit in saying "observational" before cross-sectional. Remove the word "observational".

7. What is the conclusion of your study?

Introduction

8. Line 33 - 44: these lines do not contribute significantly to your overall narrative, you can either remove or shorten them.

9. Line 57 - 59: the most common side effects after COVID-19 vaccines were not cutaneous side effects.

10. Your Introduction narrative may benefit from reflecting on similar studies findings of COVID-19 vaccine side effects.

Suggested refs:

https://www.frontiersin.org/articles/10.3389/fpubh.2022.834744/full

Materials and Methods

11. The STROBE guidelines for cross-sectional studies should be used and cited properly.
https://www.thelancet.com/journals/lancet/article/PIIS0140-6736(07)61602-X/fulltext#article_upsell

12. The STROBE checklist should be added as a supplementary file.

13. How the questionnaire used in this study was developed and validated?

14. What is the rationale for differentiating adverse events and side effects?

Results

15. How many participants were initially targeted? and excluded?

16. Table 1. Please fix this word "typification"

17. Table 1. you can combine the frequencies and percentages of each subgroup in one column to facilitate their reading.
For example 6 (50%)

Discussion

18. The Discussion section is too short.

19. The findings of this study should be discussed more deeply and also compared to previous literature.

20. The strengths and implications of this study should be added.

Sincerely,

Author Response

(The authors gave the same response as above.)

Round 2

Reviewer 1 Report

The Authors addressed all the points raised.

Author Response

We appreciate your comments for the improvement of the final writing.

Reviewer 2 Report

Dear Author(s),

Thank you for your esteemed efforts in addressing my previous comments. The manuscript has improved significantly.

The Discussion section still lacks reflections on similar studies from other countries.

Suggest refs (optional):

https://doi.org/10.3389/fpubh.2022.937794

https://doi.org/10.3390/vaccines9060673

Sincerely,

Author Response

Thank you very much for your observations, we have corrected, added, and erased everything you suggested us in order to improve the manuscript.

Reviewer commentary-Observations and suggestions

Response

The Discussion section still lacks reflections on similar studies from other countries.

We have added three more similar studies in this section.

Suggested references in relation to the original manuscript have been added